# Comprehensive Review of Mechanisms and Translational Perspectives on Programmed Cell Death in Vascular Calcification

**DOI:** 10.3390/biom15121640

**Published:** 2025-11-21

**Authors:** Yiyang Cao, Yulong Cao, Jiaheng Liu, Yifei Ye, Meixiu Jiang

**Affiliations:** 1The Queen Mary School, Jiangxi Medical College, Nanchang University, 999 Xuefu Road, Nanchang 330031, China; 2School of Pharmacy, Jiangxi Medical College, Nanchang University, 999 Xuefu Road, Nanchang 330031, China; 3Jiangxi Province Key Laboratory of Bioengineering Drugs, The National Engineering Research Center for Bioengineering Drugs and the Technologies, Institute of Translational Medicine, Jiangxi Medical College, Nanchang University, 999 Xuefu Road, Nanchang 330031, China

**Keywords:** vascular calcification, programmed cell death, molecular signal transduction, translational targets

## Abstract

Programmed cell death (PCD), a regulated cell death (RCD) subtype essential for physiological homeostasis, encompasses distinct mechanisms including apoptosis, necroptosis, autophagy, ferroptosis, cuproptosis, and pyroptosis. This evolutionarily conserved process critically regulates disease pathogenesis across degenerative disorders, malignancies, fungal infections, and vascular calcification (VC). VC manifests as pathological calcium deposition in cardiovascular tissues, impairing vascular elasticity and hemodynamics. These structural alterations elevate risks of atherosclerotic events, cerebrovascular accidents, and peripheral vascular dysfunction, while concomitantly inducing vital organ hypoperfusion and cardiac overload that predispose individuals to myocardial ischemia, left ventricular hypertrophy, and heart failure. Despite advances in delineating associated signaling networks, the molecular etiology of VC remains elusive, and effective therapeutic interventions are currently lacking. While systematically examining the pathophysiological contributions of both established and novel PCD mechanisms to VC progression, this review incorporates a discussion of cuproptosis as a novel form of PCD, which may serve as a target for atherosclerosis treatment. The inclusion of cuproptosis, alongside other PCD modalities, allows for a more integrated and updated perspective on the complex regulatory networks governing VC. Our objective is to synthesize the current understanding of how these diverse PCD pathways, both classical and emerging, collectively contribute to the disease pathogenesis and to explore the broader therapeutic potential of targeting PCD in VC.

## 1. Introduction

Vascular calcification (VC), a hallmark of aging, commonly occurs in atherosclerosis [1] and diabetes mellitus [2], posing significant health risks. This pathological process involves excessive calcium deposition in the cardiovascular system, leading to reduced arterial elasticity, impaired blood circulation, and compromised organ perfusion [3]. Associated cardiovascular complications include myocardial hypertrophy and hypertension [4]. Clinically, VC severity serves as an independent predictor for assessing cardiovascular morbidity and mortality.

The scientific community recognizes VC as a complex pathophysiological process [5]. Unlike physiological bone mineralization during remodeling, VC progression is driven by the phenotypic transformation of vascular smooth muscle cells (VSMCs) into osteoblast-like cells, a process marked by the downregulation of contractile markers and the concomitant upregulation of osteogenic transcription factors [6]. This phenotypic switching is principally orchestrated by the activation of specific osteogenic signaling pathways and transcription factors. Key among these are the Runt-related transcription factor 2 (Runx2) and Msh homeobox 2 (Msx2), which serve as master regulators of bone-forming protein expression, including Alkaline Phosphatase (ALP), Osteopontin (OPN), and Osteocalcin [7,8,9]. Established inducers of this process include dysregulated mineral metabolism (particularly hyperphosphatemia) [10], Bone Morphogenetic Protein (BMP), and Wnt/β-catenin signaling pathways [11], which collaboratively promote the deposition of hydroxyapatite crystals within the vascular wall. While VSMCs constitute the primary cellular mediators, emerging evidence suggests synergistic contributions from endothelial cells through paracrine signaling and pericytes via their multipotent differentiation potential, amplifying the calcification cascade [12,13].

VC is classified into three types based on calcification sites: intimal, medial, and valvular. Intimal calcification, strongly linked to atherosclerotic plaques, involves chondrocyte-like cells regulated by macrophage/foam cell-derived cytokines that stabilize plaques. Inflammatory stimuli and hyperphosphatemia drive VSMCs to adopt osteoblast-like phenotypes, accelerating calcification [3]. Microcalcifications may evolve into calcified nodules or bone-like structures [14], whose hazards are closely related to the pathological process. In atherosclerotic plaques, microcalcifications disrupt the integrity of the fibrous cap through local stress concentration, significantly increasing the risk of plaque rupture and acute coronary events [15]. Meanwhile, these calcified particles can activate macrophages to release pro-inflammatory factors, exacerbating the vascular inflammatory microenvironment [16]. Medial calcification arises in elastin-rich vascular walls where VSMCs undergo osteogenic transformation via osteoprogenitor cells and transcription factors (Msx2, SRY-box transcription factor 9 [Sox9], Runx2). Bone morphogenetic proteins and inflammatory mediators activate Msx2/Wnt pathways, inducing bone-associated protein expression [17,18]. This is particularly prevalent in patients with diabetes and chronic kidney disease (CKD) [19]. Hyperphosphatemia contributes to VC by inducing osteoblastic differentiation of VSMCs, leading to abnormal hydroxyapatite deposition within the vascular elastic lamina, resulting in increased arterial stiffness and systolic hypertension [20]. In lower extremity arteries, medial calcification critically impairs blood flow perfusion, amplifying amputation risks in affected populations [21]. Healthy aortic valves consist of three thin leaflets containing valve endothelial cells and fibroblast-like interstitial cells [22]. Mechanical stress-induced endothelial damage promotes lipoprotein deposition, oxidized lipid accumulation, and immune infiltration, initiating valve calcification and aortic stenosis [23].

As a multisystem pathology, VC elevates disease incidence and mortality. Limited therapeutic options exist due to non-traditional risk factors (hyperphosphatemia, hypercalcemia, thyroid dysfunction), with current strategies focusing on early intervention: dietary phosphorus control, phosphate binders, vitamin D receptor activators, and calcium-sensing receptor agonists to normalize calcium/phosphorus levels and prevent secondary hyperparathyroidism [24,25]. Osteoprotegerin (OPG) inhibits VC by blocking osteoclast activity [26], suggesting the potential for bone homeostasis-targeting drugs (bisphosphonates, denosumab). Klotho protein demonstrates anti-calcification effects, though its therapeutic potential via endogenous/exogenous supplementation requires validation [27]. Most mechanisms remain incompletely understood, necessitating further research.

Cellular homeostasis relies on balanced regulated cell death (RCD) and accidental cell death. RCD, encompassing programmed cell death (PCD) subforms like apoptosis, necroptosis, and pyroptosis, maintains physiological integrity through controlled signaling cascades [28]. PCD critically influences development (cardiac morphogenesis [29] and immunity [30]) and disease progression (degenerative disorders, cancers [31], and VC [32]). In VC, dynamic interactions among cell types, molecular mediators, and PCD pathways regulate calcification dynamics. For instance, Wang et al. showed that peptide N-acetylgalactosaminyltransferase 3 reduces VSMC apoptosis, alleviating aortic calcification in mice [33]. González-Salvatierra et al. identified sclerostin’s vascular protection via apoptosis suppression [34]. These findings highlight PCD regulation as a therapeutic nexus. Deciphering these networks may unveil novel biomarkers and therapeutic targets.

Building upon this foundation, this review aims to systematically synthesize and delineate the specific mechanistic contributions of major PCD pathways, including apoptosis, necroptosis, ferroptosis, cuproptosis, and pyroptosis, to the pathogenesis of VC. We will critically evaluate how these distinct cell death modalities interact with and drive the osteogenic differentiation of VSMCs, focusing on key signaling cross-talks and the release of critical mediators. Furthermore, another objective of this review is to explore the emerging role of cuproptosis, a newly identified PCD, in VC pathophysiology, an area that has not been comprehensively covered in previous reviews. Finally, by integrating these insights, we seek to discuss the translational potential of targeting PCD pathways as novel therapeutic strategies for mitigating VC.

## 2. PCD and VC

### 2.1. Apoptosis and VC

#### 2.1.1. Apoptosis

Apoptosis, an evolutionarily conserved PCD mechanism, eliminates damaged cells while paradoxically enabling proliferative signaling during tissue remodeling [35]. Key morphological features include cell shrinkage, chromatin margination, membrane blebbing, DNA fragmentation, and apoptotic body formation—membrane-bound vesicles phagocytosed by neighboring cells [36,37].

Apoptosis proceeds through two classical pathways: the mitochondrial (intrinsic) pathway initiated by cellular stress, and the cell surface death receptor (extrinsic) pathway [38]. In the intrinsic pathway, mitochondrial cytochrome c release combines with Apaf-1 and caspase-9 to form the apoptosome [39], recruiting caspase-3 to initiate a proteolytic cascade that amplifies death signals. Simultaneously, the second mitochondria-derived activator of caspase (Smac) neutralizes the inhibitor of apoptosis proteins (IAP) [40], apoptosis-inducing factor induces nuclear condensation [41], and endonuclease G cleaves DNA—collectively enhancing apoptotic progression. The Bcl-2 family regulates mitochondrial permeability through anti-apoptotic (Bcl-2, Bcl-xL, Mcl-1) and pro-apoptotic (Bax, Bak) proteins, together with BH3-only activators (Bid, Bim, Puma, etc.) [42]. Caspase-3 cleavage of Bcl-2 at Asp34 generates pro-apoptotic fragments that accelerate apoptosis [43]. The extrinsic pathway is activated through death receptor-ligand binding (e.g., FasL-Fas, TNF-α-TNFR1) [44], triggering receptor oligomerization and caspase-8/FAS-associated death domains recruitment to form the death-inducing signaling complex. Caspase-8 activates Bid, bridging extrinsic and intrinsic pathways to execute apoptosis [45].

#### 2.1.2. The Role of Apoptosis in VC

Apoptosis, a major form of PCD, significantly contributes to VC pathogenesis. Key mediators, including endoplasmic reticulum (ER) stress, hyperphosphatemia, growth factor deficiency, and cellular hypoxia, can trigger apoptosis-driven VC.

As a membrane-bound organelle, the ER maintains proteostasis through synthesizing, processing, and transporting cellular components. Accumulation of misfolded/unfolded proteins disrupts proteostasis, initiating ER stress. Substantial evidence confirmed that ERS regulates VC through apoptotic mechanisms, primarily via four pathways: PERK, IRE1α, ATF6, and Ca^2+^, with C/EBP-homologous protein (CHOP) as the pivotal downstream effector [46]. Notably, Liu et al. revealed that C5a-C5aR1 activates the PERK-eIF2α-ATF4 pathway by inducing ERS in CKD models, promoting VSMC osteogenic transdifferentiation [47]. This finding aligns with Masuda et al.’s demonstration of TNF-α-mediated ERS activating the same pathway in CKD-associated VC [48]. These findings highlight the therapeutic potential of targeting this pro-apoptotic axis. Recent studies further identified parathyroid hormone and 25-hydroxycholesterol as ERS-mediated VC promoters [49,50], contrasting with fibroblast growth factor-21 (FGF-21)’s protective effects [51].

The FGF family regulates cell migration, proliferation, differentiation, and tissue homeostasis through FGF/FGFR signaling [52]. Mechanistically, Shi et al. demonstrated that FGF-21 suppresses ERS-induced apoptosis by downregulating GRP78, CHOP, and caspase-12 expression, evidenced by reduced TUNEL-positive cells in VC models [53]. This establishes apoptosis inhibition as a viable therapeutic strategy. Clinical observations confirmed that serum phosphorus levels critically regulate VC progression in CKD patients. Hyperphosphatemia induces VSMC calcification through multiple mechanisms: releasing calcifying vesicles, upregulating osteogenic markers (BMP2), and activating pro-apoptotic pathways [54,55]. The OPG/Klotho axis exerts regulatory control [56], while Gas6/Axl/Akt signaling emerges as central to phosphate-induced calcification. Son et al. identified statins attenuate VSMC apoptosis/calcification by enhancing Gas6/Axl expression and Akt/Bcl2/Bad phosphorylation, thereby suppressing caspase-3 activation [57]. Complementarily, Kim et al. demonstrated α-lipoic acid’s dual antioxidant and pro-survival effects through this pathway, suggesting multi-target intervention potential [58]. In addition to this, Liu et al. found that Bone marrow mesenchymal stem cell-derived exosomes inhibited hyperphosphate-induced apoptosis and VC by carrying miR-381-3p [59]. Cui et al. provided evidence that Mitoquinone attenuates VC by inhibiting VSMC oxidative stress and apoptosis through the Keap1/Nrf2 pathway [60]. As summarized in Figure 1, apoptosis intersects with VC through direct calcification and osteogenic differentiation [61,62,63,64]. These mechanistic insights provide validated therapeutic targets and multimodal intervention strategies for VC management.

### 2.2. Necroptosis and VC

#### 2.2.1. Necroptosis

Necroptosis, a caspase-independent programmed cell death pathway, is initiated through apoptosis inhibition. Its molecular mechanism involves receptor-interacting protein kinase 3 (RIPK3) -mediated phosphorylation of mixed lineage kinase domain-like protein (MLKL) [65,66]. Phosphorylated MLKL forms plasma membrane pore complexes that release damage-associated molecular patterns (DAMP), induce cell swelling, and cause membrane rupture. Characteristic morphological features include organelle swelling, membrane blebbing, moderate chromatin condensation, and cytoplasmic/nuclear disintegration [67].

#### 2.2.2. The Role of Necroptosis in VC

Emerging evidence suggests necroptosis influences VC through multiple pathways, though mechanistic studies remain limited. As a programmed necrosis pathway, necroptosis contributes to cardiovascular pathologies, including myocardial infarction and atherosclerosis [68]. DAMPs, such as calprotectin, amplify VC-associated inflammation by stimulating oxidative stress and cytokine production [69]. Amaya-Garrido et al. demonstrated serum calprotectin elevation correlates with VC severity, while pharmacological inhibitors (paquinimod, AGE/TLR4 blockers) attenuate calcification by modulating VSMC changes [70,71]. Recent studies implicated TLR4-mediated DAMP signaling in modulating IL-6, vascular endothelial growth factor, and collagen secretion, with TGF-β critically driving VSMC phenotypic switching during calcification [72]. Necroptotic cell lysis releases cellular contents that nucleate calcium phosphate crystals, a critical step in VC progression. These deposits form on calcifiable templates like collagen and cellular debris [73].

Phosphorylated RIPK3/MLKL complexes disrupt mitochondrial membranes, inducing hyperpolarization, fragmentation, and ROS overproduction—key drivers of VSMC osteogenic transformation. MLKL directly impairs mitochondrial function, while RIPK3 activates metabolic enzymes (pyruvate dehydrogenase, glutamine synthetase) to exacerbate oxidative stress [74]. Excessive mitochondrial fission promotes VSMC calcification, which Irisin inhibits via AMPK/Drp1 signaling, whereas lactate exacerbates it through NR4A1/DNA-PKcs/p53 activation [75,76]. While RIPK1 independently promotes VC via NF-κB-mediated inflammation, its mechanisms are distinct from necroptosis and warrant separate analysis.

As summarized in Figure 2, necroptosis intersects with VC through inflammatory amplification, mitochondrial dysfunction, and mineralization nucleation [77,78,79]. Elucidating these mechanisms could yield targeted therapies to mitigate cardiovascular calcification.

### 2.3. Autophagy and VC

#### 2.3.1. Autophagy

Autophagy, a PCD mechanism, maintains cellular homeostasis by selectively degrading damaged organelles, harmful proteins, and intracellular pathogens through lysosomal recycling [80]. This process is evolutionarily conserved and controlled by autophagy-related genes (Atg) [81]. When mTOR activity is suppressed, the Unc-51-like kinase complex becomes activated and stimulates the class III PI3K complex (composed of Beclin1, Atg14L, hVps34, and p150). Once activated, this PI3K complex generates phosphatidylinositol-3-phosphate (PI3P) at pre-autophagosomal membranes, recruiting Atg proteins through hierarchical assembly mechanisms [82]. Among various forms of autophagy, mitophagy represents a selective process targeting mitochondria [83].

#### 2.3.2. The Role of Autophagy in VC

Various studies have identified that the activity of cell autophagy fluctuates with the development of VC, indicating a mechanistic link between these processes. As Atg proteins critically regulate autophagy, multiple studies have explored their roles in VC pathogenesis and therapeutic potential. For example, metformin suppresses VSMC calcification by enhancing autophagic flux to degrade Runx2, whereas Atg3 silencing increases nuclear Runx2 accumulation, exacerbating VC [84]. Peng et al. further demonstrated that estrogen inhibits VSMC osteogenic differentiation through autophagy activation, evidenced by upregulated LC3I/II and Atg5 expression [85].

Different metal ions bridge autophagy and VC. Calcium signaling modulates autophagy through multiple pathways: IP3R/beclin1 interactions, CaMKKβ-AMPK-mTOR signaling, mitochondrial Ca^2+^ uptake, and lysosomal regulation. Notably, thonningianin A attenuates VC by activating L-type calcium channels to induce autophagy [86]. Paradoxically, another research showed that low potassium environment aggravates VC via elevated calcium level-induced autophagy [87], demonstrating dual functions of autophagy and intracellular calcium towards VC. Concurrently, magnesium also exerts anti-calcification effects via the Erk/autophagy pathway, protecting extracellular matrix integrity [88].

Non-coding RNAs are also an emphasis of the latest research. The lncRNA CAIF inhibits p53-mediated myocardin transcription while suppressing autophagy, accelerating VC by impairing SOX9 antagonism [89,90]. Conversely, small nucleolar RNA host gene 1 reduces VSMC calcification under high glucose by stabilizing Bhlhe40 mRNA to suppress Atg10 expression and excessive autophagy [91]. Furthermore, ectopic calcification has been reported to correlate with particular microRNAs regulating the epigenome. miRNAs like miR-30b, a regulator of SOX9, alleviate VC via mTOR pathway-mediated autophagy induction [92], whereas macrophage-derived miR-32 promotes VSMC osteogenesis by inhibiting autophagy, counteracted by Mef2d through cGMP/PKG signaling [93].

Most studies mentioned the positive facets of autophagy, whereas research also establishes the critical role of autophagy in promoting osteoblast differentiation and mineralization in several specific situations [64]. For example, under a low potassium environment, autophagy activation can exacerbate VC by enhancing VSMC osteogenesis [87], while cold exposure suppresses it [94]. However, specific mechanisms remain unclear.

In summary, autophagy serves dual roles in VC pathogenesis (inhibition via degradation of Runx2 calcium-sensing receptor [CaSR] and mtROS; aggravation via vascular osteogenic conversion and mineralization promotion) (Figure 3) [84,95,96,97]. While pro-autophagy agents hold therapeutic promises, certain pathways paradoxically aggravate calcification. Critical gaps remain in understanding autophagosome-lysosome fusion dynamics and degradation mechanisms. Elucidating these processes is essential for developing targeted VC therapies.

### 2.4. Ferroptosis and VC

#### 2.4.1. Ferroptosis

Ferroptosis, typically involving glutathione peroxidase 4 (GPX4), glutathione (GSH), and System Xc^−^ complex, is a distinct cell death modality brought on by iron overload that leads to ROS accumulation [98]. A redox imbalance between oxidant generation and antioxidant defense primarily drives ferroptosis, mediated by dysregulated redox-active enzymes [99]. Morphologically, ferroptosis, differing from other PCD, manifests as mitochondrial shrinkage with condensed membranes and reduced/absent cristae [100].

Mechanistically, ferroptosis is primarily controlled by the System Xc^−^–GSH–GPX4 axis, where System Xc^−^ (SLC7A11/SLC3A2) imports cystine for glutathione (GSH) synthesis via GCLC and GSS, and GPX4 subsequently detoxifies phospholipid hydroperoxides (PLOOH) into non-toxic lipid alcohols [101,102]. Parallel to the GPX4 axis, the NADPH/FSP1 system also suppresses ferroptosis by regenerating lipid antioxidants independently of glutathione [103]. The tumor suppressor p53 exerts dual regulatory roles in ferroptosis: promoting it via SAT1/GLS2 upregulation and SLC7A11 inhibition, while suppressing it through DPP4–NOX1 axis inhibition and CDKN1A/p21 induction [104].

#### 2.4.2. The Role of Ferroptosis in VC

Ferroptosis is induced by iron overload, leading to ROS accumulation, indicating iron is an imperative bridge connecting ferroptosis and VC. Iron serves as an indispensable mental ion in our body, regulating several important molecules like ROS and calcium phosphate deposition, which are pathophysiologically linked to cardiovascular diseases, including atherosclerosis, hypertension, aortic valve stenosis, coronary artery disease, and VC [105,106,107]. Ferritin H, an iron storage protein with iron oxidase activity and antioxidant properties, whose mRNA expression level is directly proportional to the iron concentration in the aorta, demonstrates anti-calcification effects by suppressing phosphate-induced osteogenic differentiation in SMCs [108]. Clinical investigations further revealed that hemodialysis patients with reduced serum iron and transferrin saturation exhibit elevated coronary artery calcification scores, implicating iron deficiency in calcification progression [108]. Collectively, it can be predicted that iron is a crucial regulator of VC, and a suitable concentration of iron may attenuate VC. Notably, iron overload, a principal inducer of ferroptosis, stimulated aortic calcium deposition through osteoblast differentiation mediators (Runx2, BMP-2, Msx-2, RANKL) [109]. Furthermore, 2018 research in *Nature Portfolio* indicated interleukin-24 as a critical mediator of iron-induced VSMC calcification, demonstrating the calcification capacity of interleukin-24 even in iron-deficient environments [110]. Accordingly, aiming to unearth more effective treatments for VC, research finding the optimum iron concentration in the serum of patients should be encouraged.

ROS accumulation caused by the availability of redox-active iron and loss of lipid hydroperoxide repair capacity is the direct inducer of ferroptosis [98]. Experimental models created by palmitic acid, the most prevalent long-chain saturated fatty acid in plasma, created an in vitro oxidative stress-related calcification model and caused lipid overload, which was classified to facilitate ferroptosis in VSMCs via promoting the protein expression of the extracellular matrix protein periostin and its secretion into the extracellular environment [111]. Concomitantly, PA-mediated SIRT6 suppression in VSMCs further accelerates calcification [112], while 1-Palmitoyl-2-(5′-oxo-valeroyl)-sn-glycero-3-phosphocholine (POVPC) also exacerbates osteogenic differentiation of VSMCs via ferroptosis activation [113]. These discoveries further demonstrated the close relationship between ferroptosis and VC. Additionally, SLC7A11, a component of System Xc−, plays a crucial role in ROS regulation in the ferroptosis process [114]. Recent research has demonstrated that reducing levels of transcription factor, Fra-1 (also known as FOSL1), attenuated calcification of VSMCs, through the inhibition of ferroptosis via the p53/SLC7A11 signaling pathway [115], highlighting SLC7A11 as a valuable research target. Furthermore, POVPC-induced mitochondrial ROS promotes VSMC ferroptosis [115], while calcium-dependent Nox5-generated ROS facilitates VSMC phenotypic transition—a calcification prerequisite [116]. As mentioned, ROS-induced PLOOH was also an indispensable factor in ferroptosis [98]. Currently, voluminous chemical compounds have been discovered useful to treat VC based on the mechanisms of reducing the production and accumulation of ROS, such as Wogonin, 3-Arylcoumarin, osteoprotegerin, and Intermedin [117,118,119,120].

Notably, the latest research provided more evidence about the regulatory role of ferroptosis on VC. For instance, lipocalin-2 exacerbates ferroptosis in VSMCs via NCOA4/FTH1-mediated ferritinophagy, thereby promoting VC associated with CKD [121]. The hypothalamic peptide Nesfatin-1 alleviated calcific aortic valve disease by inhibiting cell ferroptosis mediated through the GSH/GPX4 and ZIP8/SOD2 signaling axes [122]. They both indicate the great potential for further research about ferroptosis and VC.

To summarize, ferroptosis promotes VC via mineral deposition coupling and an amplification loop (Figure 4) [123,124]. Considering that iron acts as an important antioxidant to maintain cell survival, and ferroptosis is typically triggered by ROS, we can draw a prediction that ferroptosis may be a prospective therapeutic target for VC treatment.

### 2.5. Cuproptosis and VC

#### 2.5.1. Cuproptosis

Cuproptosis is a recently defined form of programmed cell death triggered by excessive copper ions (Cu^2+^) that directly bind to lipoylated proteins within the tricarboxylic acid (TCA) cycle, inducing their aberrant aggregation and destabilizing iron–sulfur (Fe–S) cluster proteins [125,126,127]. Key regulators include ferredoxin-1 (FDX1) and lipoyl synthase, which mediate the lipoylation of TCA enzymes such as dihydrolipoamide S-acetyltransferase [128,129]. Cellular copper homeostasis, maintained through importers (solute carrier family 31 member 1 [SLC31A1]) and exporters (ATPase copper transporting [ATP7] B), critically determines cuproptosis susceptibility [130]. Notably, GSH exerts protective effects by copper chelation, revealing that antioxidant capacity modulates this death pathway [131].

#### 2.5.2. The Role of Cuproptosis in VC

Emerging evidence has revealed significant connections between copper regulation and VC. Clinical observations in 2015 revealed elevated serum copper levels in hemodialysis patients with severe corneal or AAC, though mechanistic insights remained elusive [132]. Subsequent work by Wang et al. (2018) identified abnormally high plasma copper concentrations in hemodialysis patients, suggesting copper imbalance exacerbates oxidative stress and inflammation to accelerate VC progression [133]. Paradoxically, Liu and Liang’s cross-sectional study (2019) associated high dietary copper intake with reduced AAC risk, highlighting the complex duality of copper’s role in VC pathogenesis [134].

The latest research conducted by Qi et al. indicates the fact that calcification in the aortas of VitD3-treated mice is aggravated by cuproptosis induced by elesclomol, a copper transporter. Furthermore, they proved that Elabela, a recently identified peptide, prevents cuproptosis by activating the PPAR-γ/FDX1 signaling pathway and promoting copper efflux through ATP7a-mediated mechanisms, thereby mitigating VC resulting from vitamin D3 excess [135]. A close association between cuproptosis and cardiovascular disease, such as atherosclerosis, has been identified [136], indicating that undiscovered links between cuproptosis and VC exist. Patients with atherosclerosis typically experience VC, which is more prevalent and severe in those with diabetes and CKD. Concurrently, calcification can worsen atherosclerosis by causing blood vessels to stiffen and blood pressure to rise [137]. Recently, a new observational study proved that copper plays a non-negligible role in atherosclerosis [138]. With ceRNA network analyses confirming their regulatory roles, three cuproptosis-related genes (SLC31A1, SLC31A2, SOD1) can serve as diagnostic biomarkers for atherosclerosis [139]. Additionally, using similar strategies, another research showed that cuproptosis-specific expressed genes, FDX1 and GLS, were also important atherosclerosis regulators [140]. These findings demonstrated that cuproptosis is a crucial regulator of atherosclerosis, and VC is a high possibility.

Mechanistically, the copper transporter CTR1 (encoded by SLC31A1, which can also be inhibited by Elabela [135]) governs intracellular copper balance and activates pro-calcification pathways, including MAPK/ERK1/2 and PI3K/Akt/mTOR signaling [141,142,143]. GSH, a molecule closely related to ferroptosis, protects cells from the cytotoxic effects of copper by forming a complex with excessive copper ions through its mercaptan group, while the deficiency of GSH could be a direct inducer of cuproptosis [144,145]. Several studies have identified that the depletion of GSH, which can be induced by the lower expression of GSH synthetase, significantly promoted the calcification of VSMCs under osteogenic conditions by the SLC7A11/GSH/GPX4 axis [114,146]. Notably, GSH also plays a crucial role in the inflammatory response, regulating inflammation by maintaining redox balance through NF-κB and AP-1 signaling regulation [147]. Copper overload also induces mitochondrial ROS generation, which promotes VSMC osteogenic differentiation through BMP-2, Wnt, and NF-κB pathways while impairing redox-sensitive inflammatory regulation [141,148]. They are all evidence supporting that cuproptosis may be a novel inducer of VC development.

Although direct cuproptosis-VC investigations are limited, current findings underscore the harm caused by cuproptosis to VC, potentially via osteogenic signaling driven by mitochondrial proteotoxic stress and direct activation of pro-calcific kinases (Figure 5) [135]. Future research should prioritize two targets: elucidating spatiotemporal expression patterns of cuproptosis-related genes in VC microenvironments and deciphering crosstalk between copper regulatory networks and calcification signaling cascades.

### 2.6. Pyroptosis and VC

#### 2.6.1. Pyroptosis

Pyroptosis constitutes a programmed cell death modality characterized by distinct morphological changes, primarily triggered by extracellular/intracellular homeostasis disruption during innate immune responses [28]. This caspase-dependent process manifests through two principal pathways. In the classical Pathway, Caspase-1 activation via inflammasomes such as NOD-like receptor pyrin domain containing 3 (NLRP3) promotes recruitment of the adaptor ASC and procaspase-1, leading to caspase-1 activation. Activated caspase-1 then cleaves pro-IL-1β and pro-IL-18 into their mature forms and simultaneously cleaves gasdermin D (GSDMD), releasing its N-terminal fragment (GSDMD-NT) that forms membrane pores and drives pyroptotic lysis [149,150,151,152,153,154,155]. The non-classical pathway is mediated by caspase-4/5 (human) or caspase-11 (murine), which directly cleaves GSDMD to initiate pore formation. These events also secondarily activate NLRP3 inflammasomes, linking the two pathways [156,157].

#### 2.6.2. The Role of Pyroptosis in VC

Emerging evidence establishes pyroptosis as a critical regulator of VC. Studies demonstrated that phosphate overload activated the NLRP3-caspase-1-pyroptosis axis in VSMCs, directly driving calcium deposition [158]. This pathway identifies NLRP3 and caspase-1 as potential therapeutic targets for VC intervention. The NLRP3 inflammasome, a crucial component of innate immunity, mediates VC progression through dual mechanisms: activating caspase-1 to initiate pyroptosis and IL-1β/IL-18 release during cellular stress [159] and promoting osteoclastic bone resorption that exacerbates vascular mineralization [160]. Pharmacological inhibition of NLRP3 may therefore attenuate both inflammatory cascades and bone-vascular axis dysregulation in VC.

Current research found that serum GSDMD levels were strikingly higher in end-stage kidney disease patients with moderate-to-severe VC, suggesting its utility as a clinical biomarker [161]. Mechanistically, caspase-1 cleaves GSDMD to generate GSDMD-NT that oligomerizes at cell membranes, forming lytic pores that execute pyroptosis and pro-calcific cytokine release [162]. Similarly, the gasdermin family member GSDME modulates cellular fate determination—while low expression permits apoptosis, sufficient GSDME converts cell death to pyroptosis through caspase-dependent cleavage, amplifying inflammatory responses in cardiovascular pathologies including atherosclerosis and VC [163]. Additionally, experimental models confirmed that VSMC-specific GSDME knockdown reduces macrophage infiltration, RAGE signaling, and calcification in murine.

VC [164]. Figure 6 illustrates this regulatory network, where activated pyroptosis drives VC through pore formation and pro-calcific factor release [165]. Their clinical correlations position these mediators as priority targets for translational VC research.

## 3. Clinical Perspectives

VC management employs severity-stratified therapeutic strategies. Intravascular ultrasound remains the gold standard for calcified lesion assessment [166], complemented by optical coherence tomography for detailed calcium characterization [167]. These imaging modalities enable precise calcium quantification and stent expansion prediction for pre-procedural planning. Advanced interventional techniques, including rotational atherectomy, orbital atherectomy, and intravascular lithotripsy, demonstrate efficacy in modifying calcific deposits and optimizing stent deployment for complex coronary lesions [168,169]. However, the potential drugs treating VC remain ambiguous. Emerging research identifies PCD pathways as novel therapeutic targets, with recent evidence revealing their regulatory potential in VC progression, suggesting future precision medicine approaches could combine mechanical modification with biological pathway modulation. Table 1 presents the most valuable emerging drugs targeting PCD pathways, which show the ability to attenuate VC [24,170,171,172,173,174,175,176,177,178,179,180,181,182].

## 4. Conclusions

PCD, encompassing apoptosis, necroptosis, autophagy, ferroptosis, cuproptosis, and pyroptosis, exerts a substantial influence on the pathogenesis of multiple diseases. VC constitutes a pivotal pathogenic mechanism underlying cardiovascular disorders. Although increasing evidence suggests that PCD critically regulates VC progression, a comprehensive review that systematically constructs the PCD-related regulatory network and identifies therapeutic targets remains essential for translational research.

Previous reviews have outlined the regulatory functions of PCD in VC pathogenesis and highlighted the therapeutic potential of targeting PCD in VC treatment [32,183]. Building upon these findings, the present review updates recently discovered molecular pathways and, for the first time, explores the potential involvement of cuproptosis—a newly characterized form of PCD—in VC regulation. Notably, inhibition of cuproptosis through Elabela-mediated activation of the PPAR-γ/FDX1 signaling axis has been shown to protect against arterial calcification [135]. However, further clarification of how cuproptosis regulates osteogenic conversion of VSMCs is warranted. This review emphasizes the potential of cuproptosis in VC through its mechanistic involvement, interaction with other cell death pathways, association with VC-related diseases, and its therapeutic implications, highlighting cuproptosis as a promising target for future research and intervention.

It is noteworthy that although all forms of PCD ultimately promote VC, they do so through distinct mechanisms. Apoptosis primarily contributes by generating apoptotic bodies that serve as physical “seeds” for calcium-phosphate crystal deposition [61], coupled with the direct upregulation of osteogenic genes by the ER stress downstream factor CHOP [62]. At the same time, lytic forms of death, such as necroptosis and pyroptosis, drive calcification by releasing DAMPs and pro-inflammatory cytokines (IL-1β, IL-18), thereby creating a potent inflammatory microenvironment [77,165]. Ferroptosis and cuproptosis rely on iron-dependent lipid peroxidation and copper-induced mitochondrial proteotoxic stress, respectively, with their core mechanisms centering on disordered metal ion metabolism and intense oxidative stress [123,135]. Autophagy plays the most unique role, exhibiting a dual regulatory function where the net effect depends on the integrity of the autophagic flux and the specific substrates degraded [84,95]. These fundamental differences underscore the necessity for highly specific therapeutic strategies tailored to VC subtypes dominated by different PCD pathways.

Moreover, emerging molecular targets and bioactive compounds that mitigate VC via PCD modulation were summarized in the current review, reinforcing the central role of PCD in VC regulation and offering valuable clues for translational studies. Nevertheless, critical knowledge gaps persist in delineating the mechanistic interplay between PCD and VC. Establishing definitive causal relationships remains difficult despite growing identification of upstream and downstream components. Additionally, most studies have been limited to in vitro or animal models. Thus, human-based investigations are urgently needed to validate the clinical relevance of PCD-mediated mechanisms in VC.

Addressing these challenges through advanced molecular and translational studies may pave the way for novel therapeutic strategies. A systematic and integrative understanding of PCD-driven VC mechanisms will ultimately facilitate the development of precise, mechanism-based interventions for cardiovascular calcification.

## Figures and Tables

**Figure 1 biomolecules-15-01640-f001:**
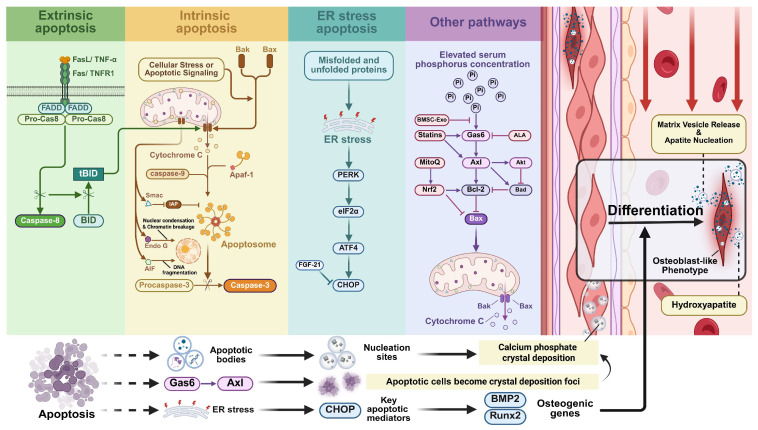
The regulatory network of apoptosis in vascular calcification (VC). Apoptosis is initiated through three major signaling cascades: extrinsic, intrinsic, and endoplasmic reticulum (ER) stress pathways. These pathways converge on effector caspases and the formation of apoptotic bodies. Apoptosis contributes to VC through two primary mechanisms: (i) Direct calcification: Apoptotic bodies serve as nucleation sites for calcium and phosphate crystal deposition. (ii) Osteogenic differentiation: Key apoptotic mediators, particularly C/EBP-homologous protein (CHOP) downstream of ER stress, directly transactivate osteogenic genes (Runx2, BMP2), promoting vascular smooth muscle cells (VSMCs) transdifferentiation into an osteoblast-like phenotype. Additionally, the Gas6/Axl pathway mediates apoptotic cells to become foci for calcium phosphate crystal deposition.

**Figure 2 biomolecules-15-01640-f002:**
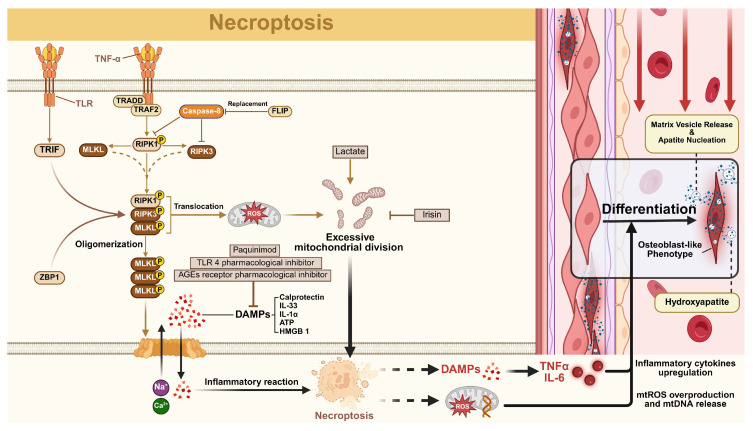
The regulatory network of necroptosis in vascular calcification (VC). Necroptosis involves the sequential assembly of the core necrosome complex, which results in the membrane translocation and pore formation. Necroptosis drives VC progression through two primary mechanisms: (i) Mitochondrial dysfunction: Necrosome activation impairs mitochondria, leading to the generation of reactive oxygen species (ROS) and metabolic changes that drive vascular smooth muscle cells (VSMCs) osteogenic transformation; (ii) Inflammatory amplification: Membrane rupture causes the release of damage-associated molecular patterns (DAMPs), establishing a robust pro-inflammatory microenvironment rich in cytokines that promotes osteogenic transdifferentiation. Various alternative triggers and modulatory factors are also included.

**Figure 3 biomolecules-15-01640-f003:**
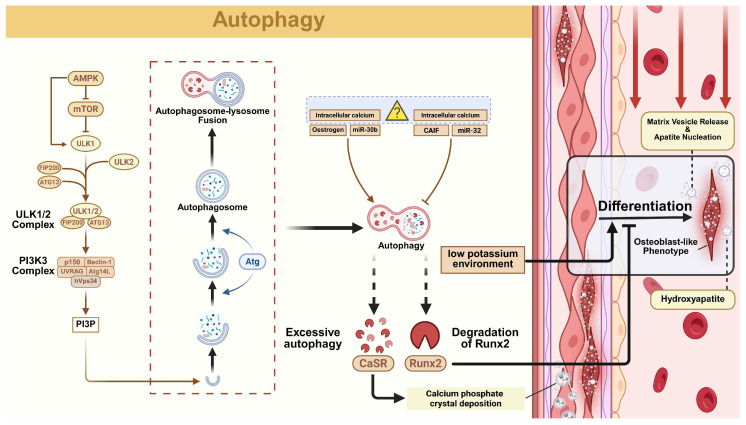
The regulatory network of autophagy in vascular calcification (VC). Autophagy is a sequential cellular process involving the formation, elongation, and closure of the autophagosome, culminating in lysosomal degradation. The autophagic process is tightly modulated by various upstream regulators, including intracellular ion concentrations, non-coding RNAs, hormonal factors, and diverse environmental cues. Autophagy directly regulates VC through the degradation of key pro-calcific factors. The autophagic degradation of Runx2 represents a primary protective mechanism by suppressing osteogenic gene expression. Conversely, excessive autophagy may promote calcification via degradation of the calcium-sensing receptor (CaSR), disrupting calcium homeostasis. Excessive autophagy also promotes VC via facilitating vascular osteogenic conversion and mineralization, but the underlying mechanisms remain unclear.

**Figure 4 biomolecules-15-01640-f004:**
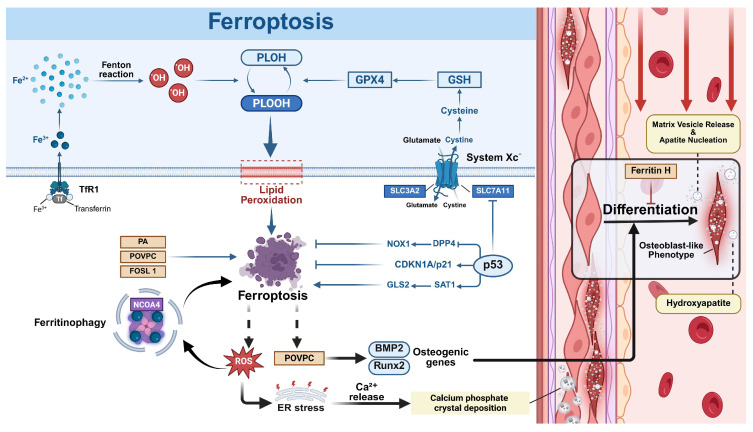
The regulatory network of ferroptosis in vascular calcification (VC). Ferroptosis is an iron-dependent form of regulated cell death, marked by the accumulation of lipid peroxides and reactive oxygen species (ROS). Ferroptosis directly promotes VC through the following downstream mechanisms: (i) Mineral deposition coupling: Ferroptosis-generated ROS induces endoplasmic reticulum (ER) stress, triggering Ca^2+^ release that facilitates calcium-phosphate deposition and directly initiates mineralization; (ii) Amplification loop: Ferroptosis-associated oxidative stress triggers nuclear receptor coactivator 4 (NCOA4)-mediated ferritinophagy, which degrades iron storage proteins and increases labile iron, thereby amplifying ferroptosis and its pro-calcific effects; (iii) 1-Palmitoyl-2-(5′-oxo-valeroyl)-sn-glycero-3-phosphocholine (POVPC) upregulates the expression of Runx2 and BMP2, inducing vascular smooth muscle cells (VSMCs) osteogenic transdifferentiation.

**Figure 5 biomolecules-15-01640-f005:**
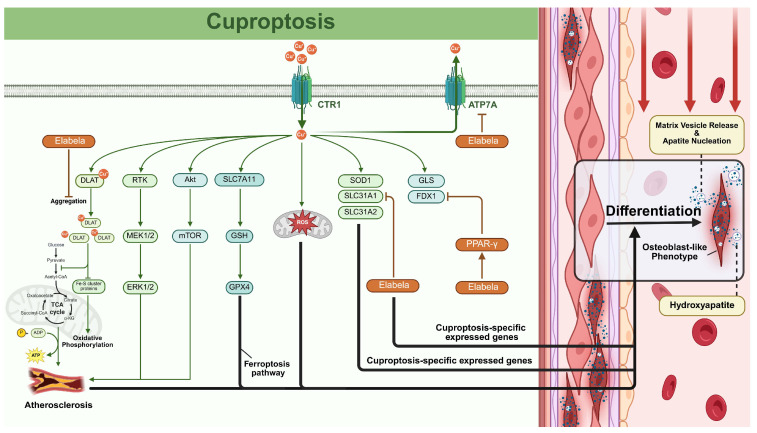
The regulatory network of cuproptosis in vascular calcification (VC). The cuproptosis and VC are linked by cuprooptosis-specific and related expressed genes, atherosclerosis, ferroptosis, and reactive oxygen species (ROS). Copper ions enter cells via copper transporter 1 (CTR1) and can activate multiple signaling pathways that lead to cuproptosis. Cuproptosis directly contributes to VC through two primary downstream pathways: (i) Mitochondrial proteotoxic stress driving osteogenic signaling: Copper-induced aggregation of lipoylated proteins and Fe-S cluster destabilization trigger mitochondrial proteotoxic stress. This results in a burst of mitochondrial ROS, which directly activates pro-osteogenic pathways (BMP-2); (ii) Direct activation of pro-calcific kinases: Intracellular copper ions directly activate key signaling cascades, including MAPK/ERK1/2 and PI3K/Akt/mTOR, which are known to phosphorylate and activate transcription factors driving the osteogenic transition of vascular smooth muscle cells (VSMCs).

**Figure 6 biomolecules-15-01640-f006:**
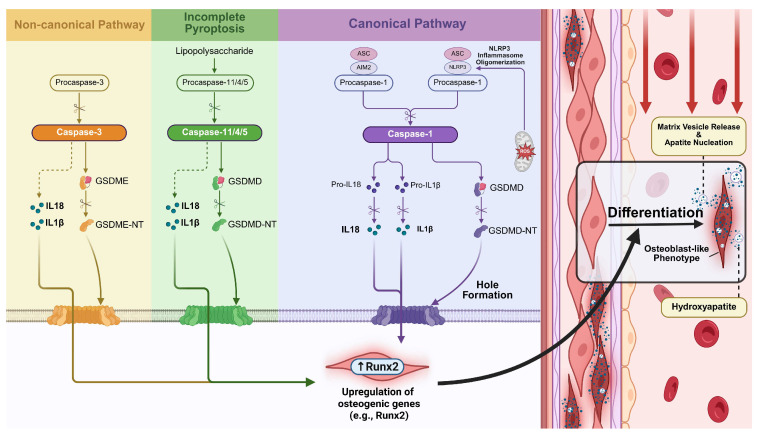
The regulatory network of pyroptosis in vascular calcification (VC). Pyroptosis is an inflammatory lytic cell death initiated by multiple convergence points: the canonical pathway, the incomplete pyroptosis pathway, and a distinct non-canonical pathway. All three cascades culminate in the cleavage of a gasdermin family protein, generating pore-forming fragments that induce cell lysis and the release of copious inflammatory cytokines. Pyroptosis directly drives VC through the downstream pathway: pore formation and pro-calcific factor release. Activated inflammatory caspases cleave GSDMD/GSDME, generating N-terminal fragments that oligomerize to form plasma membrane pores. This leads to the release of mature IL-1β and IL-18, which act directly on vascular smooth muscle cells (VSMCs), activating signaling pathways that directly upregulate the expression of key osteogenic master genes such as Runx2, thereby initiating the calcification program.

**Table 1 biomolecules-15-01640-t001:** Summarization of emerging drugs focusing on PCD and their corresponding targets.

Targeted PCD	Name	Molecular Target
Apoptosis	RAS inhibitors	α-Klotho protein
	Statins	α-Klotho protein
	mTOR inhibitors	α-Klotho protein
	Vitamin D	α-Klotho protein
	Pentophylline	α-Klotho protein
	Vitamin K2	Gas6
	Sirt1 activators	Sirtuin1
	Vitamin D	Sirtuin1
	Osteoprotegerin	RANKL
Necroptosis	GDC-8264	RIP 1
	SIR2446M	RIPK 1
	GFH312	RIPK 1
	DNL104	RIPK 1
	GSK2982772	RIPK 1
Autophagy	Irisin	NLRP3
	β-hydroxybutyric acid	LC3B
	Bavachin	mTOR
	Calycosin	mTOR
	FPS	Not discovered yet
	Atorvastatin	β-catenin
	Oestrogen	Erα
	Ghrelin	AMPK
	Exendin 4	mTOR
	Melatonin	Optic Atrophy 1
Ferroptosis	Metformin	POSTN
	DNA polymerase Gamma	P53
	PHGDH	P53
	OTC	GSH
	Oleoylethanolamide	PPARα
Cuproptosis	Elabela	PPAR-γ
Pyroptosis	Puerarin	NLRP3
	IL-38	NLRP3
	Empagliflozin	NLRP3
	Canagliflozin	NLRP3
	Sinomenine	NLRP3
	Genistein	NLRP3
	VX-765	Caspase 1
	Ligustrazine	Caspase 3

Gas6 = Growth Arrest-Specific Gene 6; RANKL = Receptor Activator for Nuclear Factor-κ B Ligand; RIP 1 = Receptor-Interacting Protein 1; FPS = Polysaccharide from Fuzi; NLRP3 = NOD-like Receptor Pyrin Domain Containing 3; LC3B = Microtubule-Associated Protein 1A/1B-Light Chain 3B; AMPK = Adenosine 5′-Monophosphate-Activated Protein Kinase; mTOR = Mammalian Target of Rapamycin; POSTN = Extracellular Matrix Protein Periostin; PHGDH = Phosphoglycerate Dehydrogenase; OTC = 2-Oxothiazolidine-4-Carboxylic Acid; SLC7A11 = Solute Carrier Family 7 Member 11; p53 = Tumor Protein 53; GSH = glutathione; PPARα = Peroxisome Proliferator-Activated Receptor α; GSDME = Gasdermin E; IL-38 = Interleukin-38.

## Data Availability

No new data were created or analyzed in this study.

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
