# Peer review of "Comprehensive Review of Mechanisms and Translational Perspectives on Programmed Cell Death in Vascular Calcification"

_biomolecules, 2025, doi:10.3390/biom15121640_

Round 1

Reviewer 1 Report

Comments and Suggestions for Authors

I have been pleased to review the manuscript by Cao et al., “Comprehensive Review of Mechanisms and Translational Perspectives Programmed Cell Death in Vascular Calcification”.

The review article described by the authors is of a basic scientific nature and has an impact on future knowledge transfer to translational research on programmed cell death in vascular calcification. This makes it very valuable and necessary, considering the impact vascular calcification has on clinical application in various pathologies.

It is a complex topic, however, the subtopics that participate in the main process are addressed in depth: Apoptosis, necroptosis, autophagy, ferroptosis, and cuproptosis. The figures presented in the review help the reader follow the processes described. Furthermore, they are well designed and clearly identify the involvement of the different processes (apoptosis, necroptosis, autophagy, ferroptosis, and cuproptosis) in vascular calcification.

I have some recommendations for the authors to improve the quality of the review:

- It's necessary to add at least one objective to the manuscript. This applies to the introduction section as well as the abstract.

- The discussion section, lines 653 - 679 is repeated in the conclusions section 681 - 707. Please join both sections (discussion and conclusions) and leave the same text.

- The authors could describe the strengths and weaknesses of their work in the discussion section, comparing this with references on the topic.

Author Response

Comments 1: It's necessary to add at least one objective to the manuscript. This applies to the introduction section as well as the abstract.

Response 1: We appreciate the reviewer’s valuable suggestion regarding the inclusion of study objectives. In response, we have added a clear statement of the research objective in both the Abstract and the Introduction sections. This addition helps to clarify the focus and purpose of the study for readers.

Below are the added paragraphs:

Abstract part (line 14-33): While systematically examining the pathophysiological contributions of both established and novel PCD mechanisms to VC progression, this review incorporates a discussion of cuproptosis as a novel form of PCD, which may serve as a target for atherosclerosis treatment. The inclusion of cuproptosis, alongside other PCD modalities, allows for a more integrated and updated perspective on the complex regulatory networks governing VC. Our objective is to synthesize the current understanding of how these diverse PCD pathways, both classical and emerging, collectively contribute to the disease pathogenesis and to explore the broader therapeutic potential of targeting PCD in VC.

Introduction part (line 106-115): Building upon this foundation, this review aims to systematically synthesize and delineate the specific mechanistic contributions of major PCD pathways, including apoptosis, necroptosis, pyroptosis, and ferroptosis, to the pathogenesis of vascular calcification. We will critically evaluate how these distinct cell death modalities interact with and drive the osteogenic differentiation of VSMCs, focusing on key signaling cross-talks and the release of critical mediators. Furthermore, another objective of this review is to explore the emerging role of cuproptosis, a newly identified PCD, in VC pathophysiology, an area that has not been comprehensively covered in previous reviews. Finally, by integrating these insights, we seek to discuss the translational potential of targeting PCD pathways as novel therapeutic strategies for mitigating VC.

Comments 2: The discussion section, lines 653 - 679 is repeated in the conclusions section 681 - 707. Please join both sections (discussion and conclusions) and leave the same text. The authors could describe the strengths and weaknesses of their work in the discussion section, comparing this with references on the topic.

Response 2: Thank you for pointing out the repetitive sections in the manuscript. We appreciate your careful attention to the structure and depth of our discussion. In response, we have removed the redundant paragraphs and expanded the discussion to include a comparison with previous research, as well as a more detailed analysis of the strengths and limitations of the current review.

Two previous reviews and one original article have been durther discussed. One previous review article published in 2022 summarized the molecular mechanisms of how PCD regulates the pathogenesis of atherosclerosis and vascular calcification (doi:10.1038/s41419-022-04923-5). Another mini-review article published recently summarized the evidence that through PCD regulation, VC progression may be attenuated (doi:10.3389/fcvm.2025.1549857).

Compared to the previous two reviews, considering the emerging new findings and lack of comprehensiveness, we updated and systemically constructed molecular networks between VC and PCD. Additionally, we summarized the potential bioactive compounds attenuating VC via PCD regulation and their targets, which may serve as valuable clues and guidance for further translational research. Notably, we proposed that cuproptosis, a newly defined form of PCD, may also be imperative in VC pathogenesis, whose potential hasn’t been summarized elsewhere.

Several critical research gaps remain to be addressed. First, the precise mechanisms by which various forms of programmed cell death, particularly cuproptosis, regulate the osteogenic conversion of VSMCs and contribute to VC progression are still unclear (the only direct evidence: doi: 10.1186/s10020-024-00997-3). Second, definitive causal relationships between PCD processes and VC development have yet to be firmly established. Furthermore, most current findings are derived from in vitro or animal models, highlighting the urgent need for human-based studies to confirm clinical relevance.

Based on your valuable recommendations and the discussions above, the conclusion section has been revised as follows (IN BOLD REDNESS), with the strengths and limitations noted.

(line: 525-535)

Previous reviews have outlined the regulatory functions of PCD in VC pathogenesis and highlighted the therapeutic potential of targeting PCD in VC treatment (1, 2). Building upon these findings, the present review updates recently discovered molecular pathways and, for the first time, explores the potential involvement of cuproptosis newly characterized form of PCD-in VC regulation. Notably, inhibition of cuproptosis through Elabela-mediated activation of the PPAR-γ/FDX1 signaling axis has been shown to protect against arterial calcification (3). However, further clarification of how cuproptosis regulates osteogenic conversion of VSMC is warranted. This review emphasizes the potential of cuproptosis in VC through its mechanistic involvement, interaction with other cell death pathways, association with VC-related diseases and its therapeutic implications, highlighting cuproptosis as a promising target for future research and intervention.

(line: 550-561)

Moreover, emerging molecular targets and bioactive compounds that mitigate VC via PCD modulation were summarized in the current review, reinforcing the central role of PCD in VC regulation and offering valuable clues for translational studies. Nevertheless, critical knowledge gaps persist in delineating the mechanistic interplay between PCD and VC. Establishing definitive causal relationships remains difficult despite growing identification of upstream and downstream components. Additionally, most studies have been limited to in vitro or animal models. Thus, human-based investigations are urgently needed to validate the clinical relevance of PCD-mediated mechanisms in VC.

Addressing these challenges through advanced molecular and translational studies may pave the way for novel therapeutic strategies. A systematic and integrative understanding of PCD-driven VC mechanisms will ultimately facilitate the development of precise, mechanism-based interventions for cardiovascular calcification.

Reference:

  1. Hu, J.; Pan, D.; Li, G.; Chen, K.; Hu, X. Regulation of programmed cell death by Brd4. Cell Death Dis 2022, 13, 1059, doi:10.1038/s41419-022-05505-1.
  2. Zheng, J.; Lin, Z.; Zhong, X.; Liu, J. The relationship between programmed cell death and vascular calcification. Front Cardiovasc Med 2025, 12, 1549857, doi:10.3389/fcvm.2025.1549857. Salthouse TN, Matlaga BF. Significance of cellular enzyme activity at nonabsorbable suture implant sites: silk, polyester, and polypropylene. J Surg Res 1975; 19:127-132.
  3. Qi, R.Q.; Chen, Y.F.; Cheng, J.; Song, J.W.; Chen, Y.H.; Wang, S.Y.; Liu, Y.; Yan, K.X.; Liu, X.Y.; Li, J.; et al. Elabela alleviates cuproptosis and vascular calcification in vitaminD3- overloaded mice via regulation of the PPAR-γ /FDX1 signaling. Mol Med 2024, 30, 223, doi:10.1186/s10020-024-00997-3.

Reviewer 2 Report

Comments and Suggestions for Authors

In this very interesting review by Meixiu Jiang and co-workers, the authors have emphasized the mechanisms, and translational perspectives programmed cell death in vascular calcification. The Programmed cell death addressed include apoptosis, necroptosis, autophagy, ferroptosis, cuproptosis, and pyroptosis.

The manuscript is well written, and the figures are certainly a strong point, as they clearly summarize each section. The work is significant, and scientifically sound, addressing an important topic that will be of interest to the readership of the journal.

Overall, I thus have only a few comments, which should help the authors enhance the clarity and overall presentation of their work.

Diffentiation must be changed to Differentiation (see figures 1‒6).

L248-252: the sentence has been written twice.

L312: I suppose the authors mean …metal…., not mental.

Remove section 4, as the same has been written in the conclusion section.

Author Response

Comments 1: Diffentiation must be changed to Differentiation (see figures 1‒6).

Response 1: Thank you for your attention to the detailed mistakes we have made. We appreciate the reviewer’s careful attention to detail and thank you for pointing out this spelling error. The word “Diffentiation” has been corrected to “Differentiation” in all relevant figures (Figures 1–6) in the revised manuscript.

One revised example (apoptosis) has been selected and presented below (line 208):

Comments 2: L248-252: the sentence has been written twice; L312: I suppose the authors mean …metal…., not mental.

Response 2: We sincerely appreciate the reviewer’s careful attention to these errors. We have comprehensively revised this figure legend to enhance its accuracy and clarity, ensuring that all content is precise and non-repetitive. In addition, the typographical error “mental” has been corrected to “metal” in line 254.

The corrected (line: 254) word is presented below: Different metal ions bridge autophagy and VC.

Comments 3: Remove section 4, as the same has been written in the conclusion section.

Response 3: Thank you for pointing out the repetitive sections in the manuscript. We appreciate your careful attention to the structure of our discussion and conclusion. In response, we have removed the redundant paragraphs (section 4-Discussion); instead which section 4 now becomes the "Conclusions" (line 519-561).

Reviewer 3 Report

Comments and Suggestions for Authors

Vascular calcification (VC) is a risk factor for various cardiovascular diseases, such as atherosclerosis and peripheral arterial disease; however, its precise molecular etiology remains incompletely understood. In this review, Cao et al. examine the role of programmed cell death (PCD) in VC. Overall, each section of PCD is well written and provides useful information. However, the specific mechanisms by which these PCD events contribute to VC are not discussed in sufficient depth. To improve the manuscript, the authors should address the following points.  

  1. The authors noted that many PCD-related signaling pathways are involved in VSMC osteogenic differentiation, which plays a key role in VC. Therefore, it would be helpful to include a brief overview of established VSMC osteogenic differentiation pathways in the introduction.
  2. Since the signaling pathways linking PCD to VC are more important than detailed descriptions of each PCD type, the authors are encouraged to focus on how PCD contributes to VC. The explanation of individual PCD types and their signaling pathways is already well summarized in the figure legends; thus, each PCD section in the main text could be shortened. Please note that there is substantial overlap between the figure legends and the main text.
  3. Most figures primarily illustrate PCD mechanisms, while the connection between each signaling pathway and VC or VSMC differentiation is not clearly presented.
    For example, although Figure 4 details ferroptosis, it does not show how ferroptosis induces VSMC differentiation. At a minimum, Figures 2, 3, 4, and 6 should be revised to align more closely with the format of Figure 5.
  4. In the abstract (line 25), the author stated that cuproptosis is a novel treatment target, but the main text does not clearly explain why cuproptosis is emphasized over other PCD types.
  5. Are there any differences in VC induced by each type of PCD? If so, please describe them in the Discussion.
  6. In line 303, the section title “2.3.3 The signaling pathways associated with Autophagy” could be revised to “The Role of Autophagy in VC”
  7. Please enlarge the text in Figures 1 and 2 for better readability.

Author Response

Comments 1: The authors noted that many PCD-related signaling pathways are involved in VSMC osteogenic differentiation, which plays a key role in VC. Therefore, it would be helpful to include a brief overview of established VSMC osteogenic differentiation pathways in the introduction.

Response 1: Thank you for this valuable suggestion. We agree that providing an overview of established VSMC osteogenic differentiation pathways would enhance the clarity and completeness of the introduction. Accordingly, we have now incorporated a concise overview of the key transcription factors (Runx2, Msx2), downstream osteogenic markers (ALP, OPN, Osteocalcin), and major signaling pathways (BMP, Wnt/β-catenin) that drive VSMC osteogenic differentiation into the end of the second paragraph of the Introduction (lines: 45-60). This addition provides the necessary foundational context, which we believe significantly enhances the logical flow and prepares the reader for the subsequent detailed discussions on how various PCD pathways intersect with this core mechanism in VC.

The added content has been highlighted with bold redness:

The scientific community recognizes VC as a complex pathophysiological process. Unlike physiological bone mineralization during remodeling, VC progression is driven by the phenotypic transformation of VSMCs into osteoblast-like cells, a process marked by the downregulation of contractile markers and the concomitant upregulation of osteogenic transcription factors. This phenotypic switching is principally orchestrated by the activation of specific osteogenic signaling pathways and transcription factors. Key among these are the Runt-related transcription factor 2 (Runx2) and Msx2, which serve as master regulators of bone-forming protein expression, including Alkaline Phosphatase (ALP), Osteopontin (OPN), and Osteocalcin [1-3]. Established inducers of this process include dysregulated mineral metabolism (particularly hyperphosphatemia)[4], Bone Morphogenetic Protein (BMP), and Wnt/β-catenin signaling pathways [5], which collaboratively promote the deposition of hydroxyapatite crystals within the vascular wall. While VSMCs constitute the primary cellular mediators, emerging evidence suggests synergistic contributions from endothelial cells through paracrine signaling and pericytes via their multipotent differentiation potential, amplifying the calcification cascade.

Reference:

  1. Yang, L.; Dai, R.; Wu, H.; Cai, Z.; Xie, N.; Zhang, X.; Shen, Y.; Gong, Z.; Jia, Y.; Yu, F.; et al. Unspliced XBP1 Counteracts β-Catenin to Inhibit Vascular Calcification. Circ Res 2022, 130, 213-229, doi:10.1161/circresaha.121.319745.
  2. Zhang, S.; Xie, Y.; Yan, F.; Zhang, Y.; Yang, Z.; Chen, Z.; Zhao, Y.; Huang, Z.; Cai, L.; Deng, Z. Negative pressure wound therapy improves bone regeneration by promoting osteogenic differentiation via the AMPK-ULK1-autophagy axis. Autophagy 2022, 18, 2229-2245, doi:10.1080/15548627.2021.2016231.
  3. Lee, A.E.; Choi, J.G.; Shi, S.H.; He, P.; Zhang, Q.Z.; Le, A.D. DPSC-Derived Extracellular Vesicles Promote Rat Jawbone Regeneration. J Dent Res 2023, 102, 313-321, doi:10.1177/00220345221133716.
  4. Macrì, F.; Vigorito, I.; Castiglione, S.; Faggiano, S.; Casaburo, M.; Fanotti, N.; Piacentini, L.; Vigetti, D.; Vinci, M.C.; Raucci, A. High Phosphate-Induced JAK-STAT Signalling Sustains Vascular Smooth Muscle Cell Inflammation and Limits Calcification. Biomolecules 2023, 14, doi:10.3390/biom14010029.
  5. Zhang, Y.; Zhao, Y.; Xie, Z.; Li, M.; Liu, Y.; Tu, X. Activating Wnt/β-Catenin Signaling in Osteocytes Promotes Osteogenic Differentiation of BMSCs through BMP-7. Int J Mol Sci 2022, 23, doi:10.3390/ijms232416045.

Comments 2: Since the signaling pathways linking PCD to VC are more important than detailed descriptions of each PCD type, the authors are encouraged to focus on how PCD contributes to VC. The explanation of individual PCD types and their signaling pathways is already well summarized in the figure legends; thus, each PCD section in the main text could be shortened. Please note that there is substantial overlap between the figure legends and the main text.

Response 2: We sincerely appreciate the reviewer’s insightful suggestion. In response, we have carefully revised the manuscript to emphasize the signaling pathways linking PCD to VC, rather than providing lengthy descriptions of individual PCD types. Redundant information overlapping with the figure legends has been removed, and each PCD section in the main text has been appropriately condensed to improve readability and focus. To ensure the completeness of the main text and prevent overlong legends, detailed descriptions about the upstream regulatory network have been deleted in the figure legends; instead which we elucidated the downstream connection between each signaling pathway and VC, which has been presented in the response for Reviewer 3, Comment 3. These revisions help clarify how PCD contributes to the pathogenesis of VC.

To ensure the conciseness, no new reference was added.

About apoptosis:

Shortened description of process and molecular mechanisms of apoptosis (lines: 119-138)

Apoptosis, an evolutionarily conserved PCD mechanism, eliminates damaged cells while paradoxically enabling proliferative signaling during tissue remodeling. Key morphological features include cell shrinkage, chromatin margination, membrane blebbing, DNA fragmentation, and apoptotic body formation—membrane-bound vesicles phagocytosed by neighboring cells.

Apoptosis proceeds through two classical pathways: the mitochondrial (intrinsic) pathway initiated by cellular stress, and the cell surface death receptor (extrinsic) pathway. In the intrinsic pathway, mitochondrial cytochrome c release combines with Apaf-1 and caspase-9 to form the apoptosome, recruiting caspase-3 to initiate a proteolytic cascade that amplifies death signals. Simultaneously, the second mitochondria-derived activator of caspase (Smac) neutralizes the inhibitor of apoptosis proteins (IAP), apoptosis-inducing factor induces nuclear condensation, and endonuclease G cleaves DNA - collectively enhancing apoptotic progression. The Bcl-2 family regulates mitochondrial permeability through anti-apoptotic (Bcl-2, Bcl-xL, Mcl-1) and pro-apoptotic (Bax, Bak) proteins, together with BH3-only activators (Bid, Bim, Puma, etc.). Caspase-3 cleavage of Bcl-2 at Asp34 generates pro-apoptotic fragments that accelerate apoptosis. The extrinsic pathway is activated through death receptor-ligand binding (e.g., FasL-Fas, TNF-α-TNFR1), triggering receptor oligomerization and caspase-8/FAS-associated death domains recruitment to form the death-inducing signaling complex. Caspase-8 activates Bid, bridging extrinsic and intrinsic pathways to execute apoptosis.

About necroptosis:

Shortened description of process and molecular mechanisms of necroptosis (lines: 179-186)

Necroptosis, a caspase-independent programmed cell death pathway, is initiated through apoptosis inhibition. Its molecular mechanism involves receptor-interacting protein kinase 3 (RIPK3) -mediated phosphorylation of mixed lineage kinase domain-like protein (MLKL). Phosphorylated MLKL forms plasma membrane pore complexes that release damage-associated molecular patterns (DAMP), induce cell swelling, and cause membrane rupture. Characteristic morphological features include organelle swelling, membrane blebbing, moderate chromatin condensation, and cytoplasmic/nuclear disintegration.

About autophagy:

Shortened description of process and molecular mechanisms of autophagy (lines: 236-244)

Autophagy, a PCD mechanism, maintains cellular homeostasis by selectively degrading damaged organelles, harmful proteins, and intracellular pathogens through lysosomal recycling. This process is evolutionarily conserved and controlled by autophagy-related genes (Atg). When mTOR activity is suppressed, the Unc-51-like kinase complex becomes activated and stimulates the class III PI3K complex (composed of Beclin1, Atg14L, hVps34, and p150). Once activated, this PI3K complex generates phosphatidylinositol-3-phosphate (PI3P) at pre-autophagosomal membranes, recruiting Atg proteins through hierarchical assembly mechanisms. Among various forms of autophagy, mitophagy represents a selective process targeting mitochondria.

About ferroptosis:

Shortened description of process and molecular mechanisms of ferroptosis (lines: 287-300)

Ferroptosis, typically involving glutathione peroxidase 4 (GPX4), glutathione (GSH), and System Xc− complex, is a distinct cell death modality brought on by iron overload that leads to ROS accumulation. A redox imbalance between oxidant generation and antioxidant defense primarily drives ferroptosis, mediated by dysregulated redox-active enzymes. Morphologically, ferroptosis, differing from other PCD, manifests as mitochondrial shrinkage with condensed membranes and reduced/absent cristae.

Mechanistically, ferroptosis is primarily controlled by the System Xc⁻–GSH–GPX4 axis, where System Xc⁻ (SLC7A11/SLC3A2) imports cystine for glutathione (GSH) synthesis via GCLC and GSS, and GPX4 subsequently detoxifies phospholipid hydroperoxides (PLOOH) into non-toxic lipid alcohols. Parallel to the GPX4 axis, the NADPH/FSP1 system also suppresses ferroptosis by regenerating lipid antioxidants independently of glutathione. The tumor suppressor p53 exerts dual regulatory roles in ferroptosis: promoting it via SAT1/GLS2 upregulation and SLC7A11 inhibition, while suppressing it through DPP4–NOX1 axis inhibition and CDKN1A/p21 induction.

About cuproptosis:

Shortened description of process and molecular mechanisms of cuproptosis (lines: 366-375)

Cuproptosis is a recently defined form of programmed cell death triggered by excessive copper ions (Cu²⁺) that directly bind to lipoylated proteins within the tricarboxylic acid (TCA) cycle, inducing their aberrant aggregation and destabilizing iron–sulfur (Fe–S) cluster proteins. Key regulators include ferredoxin-1 (FDX1) and lipoyl synthase, which mediate the lipoylation of TCA enzymes such as dihydrolipoamide S-acetyltransferase. Cellular copper homeostasis, maintained through importers (solute carrier family 31 member 1 [SLC31A1]) and exporters (ATPase copper transporting [ATP7] B), critically determines cuproptosis susceptibility. Notably, GSH exerts protective effects by copper chelation, revealing that antioxidant capacity modulates this death pathway.

About pyroptosis:

Shortened description of process and molecular mechanisms of pyroptosis (lines: 449-460)

Pyroptosis constitutes a programmed cell death modality characterized by distinct morphological changes, primarily triggered by extracellular/intracellular homeostasis disruption during innate immune responses. This caspase-dependent process manifests through two principal pathways. In the classical Pathway, Caspase-1 activation via inflammasomes such as NOD-like receptor pyrin domain containing 3 (NLRP3) promotes recruitment of the adaptor ASC and procaspase-1, leading to caspase-1 activation. Activated caspase-1 then cleaves pro-IL-1β and pro-IL-18 into their mature forms and simultaneously cleaves gasdermin D (GSDMD), releasing its N-terminal fragment (GSDMD-NT) that forms membrane pores and drives pyroptotic lysis. The non-classical pathway is mediated by caspase-4/5 (human) or caspase-11 (murine), which directly cleaves GSDMD to initiate pore formation. These events also secondarily activate NLRP3 inflammasomes, linking the two pathways.

Comments 3: Most figures primarily illustrate PCD mechanisms, while the connection between each signaling pathway and VC or VSMC differentiation is not clearly presented. For example, although Figure 4 details ferroptosis, it does not show how ferroptosis induces VSMC differentiation. At a minimum, Figures 2, 3, 4, and 6 should be revised to align more closely with the format of Figure 5.

Response 3: We thank the reviewer for this constructive comment. We agree that the connections between each PCD signaling pathway and VC or VSMC osteogenic differentiation were not clearly illustrated in the original figures. In response, we have revised all the figures to better highlight the links between PCD mechanisms and their contributions to VC and VSMC differentiation.

Correspondingly, we have also added explanatory statements in the figure legends to clarify these relationships. These revisions improve visual coherence and strengthen the mechanistic interpretation of our review.

About apoprosis

Refined figure legends of apoptosis (line 209-217):

Figure 1. The regulatory network of apoptosis in vascular calcification (VC). Apoptosis is initiated through three major signaling cascades: extrinsic, intrinsic, and endoplasmic reticulum (ER) stress pathways. These pathways converge on effector caspases and the formation of apoptotic bodies. Apoptosis contributes to VC through two primary mechanisms: i) Direct calcification: Apoptotic bodies serve as nucleation sites for calcium and phosphate crystal deposition. ii) Osteogenic differentiation: Key apoptotic mediators, particularly C/EBP-homologous protein (CHOP) downstream of ER stress, directly transactivate osteogenic genes (Runx2, BMP2), promoting vascular smooth muscle cells (VSMCs) transdifferentiation into an osteoblast-like phenotype. Additionally, the Gas6/Axl pathway mediates apoptotic cells to become foci for calcium phosphate crystal deposition.

About necroptosis

Refined figure legends of necroptosis (line 225-233):

Figure 2. The regulatory network of necroptosis in vascular calcification (VC). Necroptosis involves the sequential assembly of the core necrosome complex, which results in the membrane translocation and pore formation. Necroptosis drives VC progression through two primary mechanisms: i) Mitochondrial dysfunction: Necrosome activation impairs mitochondria, leading to the generation of reactive oxygen species (ROS) and metabolic changes that drive vascular smooth muscle cells (VSMCs) osteogenic transformation; ii)Inflammatory amplification: Membrane rupture causes the release of damage-associated molecular patterns (DAMPs), establishing a robust pro-inflammatory microenvironment rich in cytokines that promotes osteogenic transdifferentiation. Various alternative triggers and modulatory factors are also included.

About autophagy

Refined figure legends of autophagy (line 321-330):

Figure 3. The regulatory network of autophagy in vascular calcification (VC). Autophagy is a sequential cellular process involving the formation, elongation, and closure of the autophagosome, culminating in lysosomal degradation. The autophagic process is tightly modulated by various upstream regulators, including intracellular ion concentrations, non-coding RNAs, hormonal factors, and diverse environmental cues. Autophagy directly regulates VC through the degradation of key pro-calcific factors. The autophagic degradation of Runx2 represents a primary protective mechanism by suppressing osteogenic gene expression. Conversely, excessive autophagy may promote calcification via degradation of the calcium-sensing receptor (CaSR), disrupting calcium homeostasis. Excessive autophagy also promotes VC via facilitating vascular osteogenic conversion and mineralization, but the underlying mechanisms remain unclear.

About ferroptosis

Refined figure legends of ferroptosis (line 404-413):

Figure 4. The regulatory network of ferroptosis in vascular calcification (VC). Ferroptosis is an iron-dependent form of regulated cell death, marked by the accumulation of lipid peroxides and reactive oxygen species (ROS). Ferroptosis directly promotes VC through the following down-stream mechanisms: i) Mineral deposition coupling: Ferroptosis-generated ROS induces endo-plasmic reticulum (ER) stress, triggering Ca²⁺ release that facilitates calcium-phosphate deposi-tion and directly initiates mineralization; ii) Amplification loop: Ferroptosis-associated oxidative stress triggers nuclear receptor coactivator 4 (NCOA4)-mediated ferritinophagy, which degrades iron storage proteins and increases labile iron, thereby amplifying ferroptosis and its pro-calcific effects; iii) 1-Palmitoyl-2-(5'-oxo-valeroyl)-sn-glycero-3-phosphocholine (POVPC) upregulates the expression of Runx2 and BMP2, inducing vascular smooth muscle cells (VSMCs) osteogenic transdifferentiation.

About cuproptosis

Refined figure legends of cuproptosis (line 436-446)

Figure 5. The regulatory network of cuproptosis in vascular calcification (VC). The cuproptosis and VC are linked by cuprooptosis-specific and related expressed genes, atherosclerosis, ferroptosis, and reactive oxygen species (ROS). Copper ions enter cells via copper transporter 1 (CTR1) and can activate multiple signaling pathways that lead to cuproptosis. Cuproptosis directly contributes to VC through two primary downstream pathways: i) Mitochondrial proteotoxic stress driving osteogenic signaling: Copper-induced aggregation of lipoylated proteins and Fe-S cluster destabilization trigger mitochondrial proteotoxic stress. This results in a burst of mitochondrial ROS, which directly activates pro-osteogenic pathways (BMP-2); ii) Direct activation of pro-calcific kinases: Intracellular copper ions directly activate key signaling cascades, including MAPK/ERK1/2 and PI3K/Akt/mTOR, which are known to phosphorylate and activate transcription factors driving the osteogenic transition of vascular smooth muscle cells (VSMCs).

About pyroptosis

Refined figure legends of Pyroptosis (line 499-509):

Figure 6. The regulatory network of pyroptosis in vascular calcification (VC). Pyroptosis is an inflammatory lytic cell death initiated by multiple convergence points: the canonical pathway, the incomplete pyroptosis pathway, and a distinct non-canonical pathway. All three cascades culminate in the cleavage of a gasdermin family protein, generating pore-forming fragments that induce cell lysis and the release of copious inflammatory cytokines. Pyroptosis directly drives VC through the downstream pathway: pore formation and pro-calcific factor release. Activated in-flammatory caspases cleave GSDMD/GSDME, generating N-terminal fragments that oligomerize to form plasma membrane pores. This leads to the release of mature IL-1β and IL-18, which act directly on vascular smooth muscle cells (VSMCs), activating signaling pathways that directly upregulate the expression of key osteogenic master genes such as Runx2, thereby initiating the calcification program.

Comments 4: In the abstract (line 25), the author stated that cuproptosis is a novel treatment target, but the main text does not clearly explain why cuproptosis is emphasized over other PCD types.

Response 4: We appreciate the reviewer’s valuable comment. We would like to clarify that our intention was not to emphasize cuproptosis as being more important than other forms of PCD, but rather to highlight it as a recently identified type of cell death whose role in VC has not yet been comprehensively reviewed. Few direct evidence indicating the association between cuproptosis and VC have been identified. However, our study clarified the potential of cuproptosis from mechanistic involvement, interaction with other cell death pathways, association with VC-related diseases and its therapeutic implications. For instance, clinical evidence links elevated serum copper levels to severe VC in hemodialysis patients (1). Furthermore, interventional studies in VitD3-overload mouse models demonstrate that inducing cuproptosis with elesclomol aggravates aortic calcification, while the peptide Elabela mitigates VC by inhibiting cuproptosis through the PPAR-γ/FDX1 pathway and promoting copper export (2). Mechanistically, intracellular copper activates pro-calcification pathways like MAPK/ERK and PI3K/Akt/mTOR, and induces mitochondrial ROS that drives osteogenic differentiation via BMP-2 and NF-κB (3). The shared role of GSH depletion in both cuproptosis and ferroptosis, along with the identification of cuproptosis-related genes (e.g., FDX1, SLC31A1) as diagnostic biomarkers for atherosclerosis (a pathology intimately linked with VC), further underscores its potential relevance (4-5). To avoid any possible misunderstanding, we have revised the corresponding expression in the Abstract to adopt a more balanced tone, ensuring that cuproptosis is presented as a potential future research focus rather than a currently established therapeutic target.

Original expression: This review for the first time explores the potential of cuproptosis as novel treatment targets, while we systematically examined the pathophysiological contributions of PCD mechanisms to VC progression and corresponding drugs as well as the translational targets.

Refinded expression (line 25-33): While systematically examining the pathophysiological contributions of both established and novel PCD mechanisms to VC progression, this review incorporates a discussion of cuproptosis as a novel form of PCD, which may serve as a target for atherosclerosis treatment. The inclusion of cuproptosis, alongside other PCD modalities, allows for a more integrated and updated perspective on the complex regulatory networks governing VC. Our objective is to synthesize the current understanding of how these diverse PCD pathways, both classical and emerging, collectively contribute to the disease pathogenesis and to explore the broader therapeutic potential of targeting PCD in VC.

Reference:

  1. Sun, W.; Sun, M.; Zhang, M.; Liu, Y.; Lin, X.; Zhao, S.; Ma, L. Correlation between conjunctival and corneal calcification and cardiovascular calcification in patients undergoing maintenance hemodialysis. Hemodialysis International 2015, 19, 270-278, doi:https://doi.org/10.1111/hdi.12236.
  2. Qi, R.Q.; Chen, Y.F.; Cheng, J.; Song, J.W.; Chen, Y.H.; Wang, S.Y.; Liu, Y.; Yan, K.X.; Liu, X.Y.; Li, J.; et al. Elabela alleviates cuproptosis and vascular calcification in vitaminD3- overloaded mice via regulation of the PPAR-γ /FDX1 signaling. Mol Med 2024, 30, 223, doi:10.1186/s10020-024-00997-3.
  3. Ba, L.; Gao, J.; Chen, Y.; Qi, H.; Dong, C.; Pan, H.; Zhang, Q.; Shi, P.; Song, C.; Guan, X.; et al. Allicin attenuates pathological cardiac hypertrophy by inhibiting autophagy via activation of PI3K/Akt/mTOR and MAPK/ERK/mTOR signaling pathways. Phytomedicine 2019, 58, 152765, doi:10.1016/j.phymed.2018.11.025.
  4. Xue, Q.; Kang, R.; Klionsky, D.J.; Tang, D.; Liu, J.; Chen, X. Copper metabolism in cell death and autophagy. Autophagy 2023, 19, 2175-2195, doi:10.1080/15548627.2023.2200554.
  5. Patel, J.J.; Bourne, L.E.; Thakur, S.; Farrington, K.; Gorog, D.A.; Orriss, I.R.; Baydoun, A.R. 2-Oxothiazolidine-4-carboxylic acid inhibits vascular calcification via induction of glutathione synthesis. J Cell Physiol 2021, 236, 2696-2705, doi:10.1002/jcp.30036.

Comments 5: Are there any differences in VC induced by each type of PCD? If so, please describe them in the Discussion.

Response 5: We appreciate the reviewer’s insightful comment. Indeed, different types of PCD contribute to VC through distinct molecular mechanisms. We have added a concise description of these differences in the Discussion and further reflected this concept in the revised figures and their legend to provide an integrated overview, as shown above.

Added description (line 536-549):

It is noteworthy that although all forms of PCD ultimately promote VC, they do so through distinct mechanisms. Apoptosis primarily contributes by generating apoptotic bodies that serve as physical "seeds" for calcium-phosphate crystal deposition [1], coupled with the direct upregulation of osteogenic genes by the ER stress downstream factor CHOP [2]. In contrast, lytic forms of death, such as necroptosis and pyroptosis, drive calcification by releasing DAMPs and pro-inflammatory cytokines (IL-1β, IL-18), thereby creating a potent inflammatory microenvironment [3,4]. Ferroptosis and cu-proptosis rely on iron-dependent lipid peroxidation and copper-induced mitochondrial proteotoxic stress, respectively, with their core mechanisms centering on disordered metal ion metabolism and intense oxidative stress [5,6]. Autophagy plays the most unique role, exhibiting a dual regulatory function where the net effect depends on the integrity of the autophagic flux and the specific substrates degraded [7,8]. These fundamental differences underscore the necessity for highly specific therapeutic strategies tailored to VC subtypes dominated by different PCD pathways.

Reference:

  1. Neels, J.G.; Gollentz, C.; Chinetti, G. Macrophage death in atherosclerosis: potential role in calcification. Front Immunol 2023, 14, 1215612, doi:10.3389/fimmu.2023.1215612.
  2. Tóth, A.; Lente, G.; Csiki, D.M.; Balogh, E.; Szöőr, Á.; Nagy, B., Jr.; Jeney, V. Activation of PERK/eIF2α/ATF4/CHOP branch of endoplasmic reticulum stress response and cooperation between HIF-1α and ATF4 promotes Daprodustat-induced vascular calcification. Front Pharmacol 2024, 15, 1399248, doi:10.3389/fphar.2024.1399248.
  3. More, S.A.; Ghosh, A.; Kulkarni, O.P.; Mulay, S.R. Role of Persistent Necroinflammation in Chronic Tissue Remodelling and Organ Fibrosis. Am J Physiol Cell Physiol 2025, doi:10.1152/ajpcell.00416.2025.
  4. Ruan, H.; Zhang, H.; Feng, J.; Luo, H.; Fu, F.; Yao, S.; Zhou, C.; Zhang, Z.; Bian, Y.; Jin, H.; et al. Inhibition of Caspase-1-mediated pyroptosis promotes osteogenic differentiation, offering a therapeutic target for osteoporosis. Int Immunopharmacol 2023, 124, 110901, doi:10.1016/j.intimp.2023.110901.
  5. Qi, R.Q.; Chen, Y.F.; Cheng, J.; Song, J.W.; Chen, Y.H.; Wang, S.Y.; Liu, Y.; Yan, K.X.; Liu, X.Y.; Li, J.; et al. Elabela alleviates cuproptosis and vascular calcification in vitaminD3- overloaded mice via regulation of the PPAR-γ /FDX1 signaling. Mol Med 2024, 30, 223, doi:10.1186/s10020-024-00997-3.
  6. Zhong, P.; Li, L.; Feng, X.; Teng, C.; Cai, W.; Zheng, W.; Wei, J.; Li, X.; He, Y.; Chen, B.; et al. Neuronal ferroptosis and ferroptosis-mediated endoplasmic reticulum stress: Implications in cognitive dysfunction induced by chronic intermittent hypoxia in mice. Int Immunopharmacol 2024, 138, 112579, doi:10.1016/j.intimp.2024.112579.
  7. Phadwal, K.; Koo, E.; Jones, R.A.; Forsythe, R.O.; Tang, K.; Tang, Q.; Corcoran, B.M.; Caporali, A.; MacRae, V.E. Metformin protects against vascular calcification through the selective degradation of Runx2 by the p62 autophagy receptor. J Cell Physiol 2022, 237, 4303-4316, doi:10.1002/jcp.30887.
  8. Liu, C.J.; Cheng, C.W.; Tsai, Y.S.; Huang, H.S. Crosstalk between Renal and Vascular Calcium Signaling: The Link between Nephrolithiasis and Vascular Calcification. Int J Mol Sci 2021, 22, doi:10.3390/ijms22073590.

Comments 6: In line 303, the section title “2.3.3 The signaling pathways associated with Autophagy” could be revised to “The Role of Autophagy in VC”.

Response 6: We thank the reviewer for this helpful suggestion. To maintain consistency and uniformity across all section titles in the manuscript, we have revised the section heading to “The Role of Autophagy in VC”, aligning it with the format used for other PCD types. This change improves readability and ensures a coherent structure throughout the manuscript.

Original expression: The Signaling Pathways Associated with Autophagy

Refinded expression (line: 245): The Role of Autophagy in VC

Comments 7: Please enlarge the text in Figures 1 and 2 for better readability.

Response 7: We appreciate the reviewer’s suggestion. The text in Figures 1 and 2 has been enlarged to improve readability and ensure clarity in both print and digital formats.

Figure 1 (as shown on the next page):

Figure 2:

Round 2

Reviewer 1 Report

Comments and Suggestions for Authors

I thank the authors for the improvement of the submitted manuscript.

The authors adequately responded to each of the questions and comments. The process of cuproptosis and vascular calcification is of great relevance.

Now, the revised version of the review has the necessary scientific rigor. This work undoubtedly, contributes to the increase knowledge.

I congratulate the authors on their work. 

Reviewer 3 Report

Comments and Suggestions for Authors

The authors adequately addressed my previous comments.